# Ceftazidime/Avibactam-Resistant *Klebsiella pneumoniae* subsp. *pneumoniae* Isolates in a Tertiary Italian Hospital: Identification of a New Mutation of the Carbapenemase Type 3 (KPC-3) Gene Conferring Ceftazidime/Avibactam Resistance

**DOI:** 10.3390/microorganisms9112356

**Published:** 2021-11-15

**Authors:** Carla Fontana, Marco Favaro, Laura Campogiani, Vincenzo Malagnino, Silvia Minelli, Maria Cristina Bossa, Anna Altieri, Massimo Andreoni, Loredana Sarmati

**Affiliations:** 1Department of Experimental Medicine, “Tor Vergata” University, 00133 Rome, Italy; carla.fontana@uniroma2.it (C.F.); favaro@uniroma2.it (M.F.); 2Laboratory of Microbiology, Polyclinic of “Tor Vergata”, 00133 Rome, Italy; silvia.minelli@ptvonline.it (S.M.); mariacristina.bossa@ptvonline.it (M.C.B.); anna.altieri@ptvonline.it (A.A.); 3Clinical Infectious Diseases, Department of System Medicine, “Tor Vergata” University, 00133 Rome, Italy; lauracampg@gmail.com (L.C.); malagninovincenzo@gmail.com (V.M.); andreoni@uniroma2.it (M.A.)

**Keywords:** ceftazidime/avibactam, multi-drug resistance, antibiotic resistance, beta-lactamase inhibitors resistance, Q168L substitution

## Abstract

Several *Klebsiella pneumoniae* carpabenemase (KPC) gene mutations are associated with ceftazidime/avibactam (CAZ-AVI) resistance. Here, we describe four *Klebsiella pneumoniae* subsp. *pneumoniae* CAZ-AVI-resistant clinical isolates, collected at the University Hospital of Tor Vergata, Rome, Italy, from July 2019 to February 2020. These resistant strains were characterized as KPC-3, having the transition from cytosine to thymine (CAC-TAC) at nucleotide position 814, with histidine that replaces tyrosine (H272Y). In addition, two different types of KPC gene mutations were detected. The first one, common to three strains, was the D179Y (G532T), associated with CAZ-AVI resistance. The second mutation, found only in one strain, is a new mutation of the KPC-3 gene: a transversion from thymine to adenine (CTG-CAG) at nucleotide position 553. This mutation causes a KPC variant in which glutamine replaces leucine (Q168L). None of the isolates were detected by a rapid immunochromatographic assay for detection of carbapenemase (NG Biotech, Guipry, France) and were unable to grow on a selective chromogenic medium Carba SMART (bioMerieux, Firenze, Italy). Thus, they escaped common tests used for the prompt detection of *Klebsiella pneumoniae* KPC-producing.

## 1. Introduction

Antimicrobial resistance (AMR) is currently one of the major concerns in human health [1,2]. Gram-negative multidrug-resistant (MDR) pathogens represent a major challenge due to the increase in AMR worldwide [3,4] and the high disease burden, with increases in hospital stays, disability-adjusted life years, and mortality [1,2]. The main concern regarding gram-negative AMR includes resistance to carbapenems. A drug pipeline to tackle this threat has been implemented, with two novel β-lactam beta-lactamase inhibitor (BLBLI) combinations released in 2015: ceftazidime/avibactam and ceftolozane/tazobactam [3,5,6,7]. Ceftazidime/avibactam (CAZ-AVI) is a BLBLI combination drug active against carbapenem-resistant *Enterobacteriaceae* (CRE) [3,6,7,8,9]. Avibactam recovers the activity of ceftazidime by inhibiting class A, class C, and some class D beta-lactamases, hence overcoming resistance due to carbapenemases such as *Klebsiella pneumoniae* carbapenemase (KPC). However, the combination is not active against class B beta-lactamases, such as New-Delhi (NDM) and Verona Integron-encoded (VIM) and other metallo-beta lactamases.

In Italy, the circulation of CRE is endemic [10,11], mainly due to *Klebsiella pneumoniae* subsp. *pneumoniae*, but national strategies to address the threat have not yet been fully implemented. CAZ-AVI is frequently used, not only as salvage therapy but also as a first-line choice even in empiric treatments [8]. Since BLBLI introduction, isolation of pathogens with phenotypic resistance to CAZ-AVI has been reported, frequently *Klebsiella pneumoniae* strains, especially those harboring *bla*_KPC_ gene isolates. In 2015, the first *K. pneumoniae* subsp. *pneumoniae* isolate resistant to CAZ-AVI was described in Los Angeles, USA [12], in a patient who was not previously treated with the antibiotic. Subsequently, several other cases of CAZ-AVI resistance have been reported and led to the publication of the 2018 ECDC rapid risk assessment on ceftazidime/avibactam resistance in *Enterobacteriaceae* [13,14,15,16,17,18,19]. The emergence of CAZ-AVI-resistant pathogens is a public health threat, carrying serious consequences for patients, even if surveillance studies report overall low rates of CAZ-AVI resistance in CRE [20]. Data from clinical reports show a strong association with prior CAZ-AVI exposure and resistant strain selection in more than two-thirds of cases, highlighting a selective pressure caused by the BLBLI itself [20]. Resistance to CAZ-AVI can also emerge from the selection pressure of other antibiotics, such as carbapenems, demonstrating the pivotal role of antimicrobial stewardship programs.

Different mechanisms have been associated with CAZ-AVI resistance, such as mutation of the KPC gene associated with the loss/reduction in KPC enzyme functionality, hyperexpression of the *bla*_KPC_ gene, and finally, the loss of porins [15,21,22,23]. Some KPC subtypes, such as KPC-3 (nearly 70% KPC-producing *K pneumoniae* isolates belonged from a single dominant strain, multilocus sequence type 258 [ST258]; seven variants of the KPC have been reported (KPC 2–8) but most ST258 strains produced KPC-3) and specific *K. pneumoniae* clones, are more likely to develop resistance [13]. A series of mutations in the omega loop of KPC-3 have been documented in patients with microbiological failure to CAZ-AVI treatment, including H272Y, D179Y, V240G, T243M, and EL165-166 [21,24,25,26,27,28]. The D179Y variant, alone or in combination with other mutations, results in restoration of carbapenem in vitro activity of the *K. pneumoniae* isolates [28,29].

Here we describe the characteristics of four CAZ-AVI-resistant isolates and the identification of a new amino acid substitution in KPC-3 in a *K. pneumoniae* strain resistant to CAZ-AVI.

## 2. Materials and Methods

### 2.1. Bacterial Isolates

CAZ-AVI-resistant *K. pneumoniae* subsp. *pneumoniae* clinical isolates obtained from patients admitted to the University Hospital of Tor Vergata from July 2019 to February 2020 were included in the study. Isolates with resistance mechanisms other than KPC alone were excluded from the study population. The included isolates were named PTV (acronym derived from Policlinic Tor Vergata) and numbered sequentially.

The study was approved by the local ethics committee. Given the retrospective nature of the study, written informed consent was not necessary. The study was conducted in accordance with the principles of the Declaration of Helsinki.

### 2.2. Antimicrobial Susceptibility Testing and PCR Analysis

Antimicrobial susceptibility testing (AST) was performed using ITGN Micronaut panels (Diagnostika Gmbh, Bornheim, Germany, now company of Bruker Daltonics, MA, USA) run on MICRO MIB (Bruker Daltonics, Billerica, MA, USA) and interpreted following the European Committee on Antimicrobial Susceptibility Testing (EUCAST) clinical breakpoint v 9.0. Carba SMART selective chromogenic media (bioMerieux, Firenze, Italy) was used to screen for carbapenemase-producing *Enterobacteriaceae* (CPE). Identification of carbanemases (KPC, VIM, imipenemase [IMP], NDM, oxacillin-hydrolyzing [OXA] 48) was performed using the immunochromatographic (IC) assay NG CARBA (NG Biotech, Guipry, France) according to the manufacturer’s instructions. Carbapenemases detected by IC assay were also confirmed by a multiplex real-time PCR probe-based assay (our patent ID MI2014A000327) designed for the simultaneous detection of KPC, OXA-48, VIM, and NDM in a short time (no longer than 90 min from the extraction of DNA to detection) [30].

### 2.3. DNA Extraction from the Isolates

DNA was extracted starting from a pure culture (after overnight incubation at 37 °C in an aerobic atmosphere). One colony from a fresh culture of each isolate was suspended in 200 μL of G2TM solution (Qiagen Valencia, CA, USA), and then used to extract DNA using the EZ1 Advanced XL Tissue Kit, according to the bacterial protocol recommended by the manufacturers (Qiagen, Valencia, CA, USA). The DNA was eluted in 100 μL of elution bufferTM (Qiagen), and 1 μL was used to conduct the PCRs as well as sequence analysis.

### 2.4. Sequence Analysis

For the typing of the bla_KPC_ gene, overlapping PCR reactions were performed using the following primer pairs: F-5′CGGAACCATTCGCTAAACTC3′ and R-3′GGCGGCGTTATCACTGTATT5′; F-5′CGCCGTGCAATACAGTGATA3′and R-3′CGTTGACGCCCAATCC5′ (Goldfarb et al. [31]). Amplification products were purified using the Montage PCR Centrifugal Filter Device (Millipore Corporation, Billerica, MA, USA), and sequencing was performed by Big Dye Terminators V1.1 (Applied Biosystems, Foster City, CA, USA) and migrated with an automated sequencer (ABI Prism 310; Applied Biosystems). Sequences were aligned and compared using the National Center for Biotechnology Information database (http://www.ncbi.nlm.nih.gov, 13 November 2021).

### 2.5. Bla_KPC_ Cloning

The bla_KPC_ gene of KPC-3 with the Q168L substitution (PTV4 isolate) was amplified using the following primers: F prot (ACAGCCGTTACAGCCTCTG) of our design and R874 (Naas et al. [32]). The amplicon, purified using the QIAquick PCR Purification Kit (Qiagen Valencia, CA, USA), was inserted into the pGem Teasy Vector System II (Promega; Milan, Italy) following the manufacturer’s instruction.

The chimeric plasmid was used to transform *Escherichia coli* HB101 competent cells, included in the kit (Promega). Transformants were selected on SuperCAZ/AVI medium (Lifilchem, Roseto degli Abbruzzi, Italy) and screened by multiplex real-time PCR to confirm the presence of bla_KPC_ genes. Finally, to establish the CAZAVI-MIC, transformants were tested using a microbroth dilution test (ITGN, Biomedical service) as above reported.

## 3. Results

Seven CAZ-AVI-resistant (CAZ-AVI-R) isolates were identified in biological samples collected from patients hospitalized from 3 July 2019 to 5 February 2020 at Policlinic Tor Vergata University Hospital, Rome, Italy. Isolates were all *Klebsiella pneumoniae*, of which two expressed class B-beta-lactamases (non-KPC) and one had more than one resistance mechanism. Four isolates (57.1%) expressed only the KPC genotype, representing our final study population (Figure 1).

All patients had CAZ-AVI-sensitive *K. pneumoniae* (CAZ-AVI-S) prior to isolating a CAZ-AVI-R strain. Sample details and timing of collection are shown in Table 1. All but one CAZ-AVI-R isolates were resistant to carbapenems exhibiting an MIC for meropenem ranging from 16 to >64 mg/L. The time elapsed from the isolation of a CAZ-AVI-R in patients with previous isolates of CAZ-AVI-S ranged from 1 to 52 days. The antimicrobial susceptibility results for the four CAZ-AVI-R isolates are shown in Table 2.

All CAZ-AVI-R strains were unable to grow on the selective chromogenic medium Carba SMART (bioMerieux, Firenze, Italy). All CAZ-AVI-R isolates were investigated by immunochromatographic assay (NG CARBA Biotech, Guipry, France), and the KPC enzyme was undetectable. In contrast, molecular assays detected the presence of the bla_KPC_ gene in all strains, and the KPC-3 variants were identified by sequencing (Table 3).

For all isolates, the bla_KPC_ gene showed a mutation at position 814 that was a transition from cytosine to thymine CAC-TAC (UAC), identifying the KPC-3 variant, in which histidine replaced tyrosine (H272Y). PTV1-3 exhibited a substitution D179Y, owing to a single base mutation G532T (33). Sequence analysis of the KPC gene in the PTV4 strain showed an additional mutation at position 553, a transversion from thymine to adenine (CTG-CAG), that caused a variant in which glutamine replaced leucine (Q168L) (Table 3). The complete bla_KPC_ gene sequencing was deposited into the NCBI (accession number MT939316).

The bla_KPC_ cloning studies demonstrated that transformants carrying the Q168L mutation were resistant to CAZAVI, being the CAZAVI-MIC equal to 32 µg/mL in a microbroth dilution test (data not shown).

Figure 2 shows the 3D structure of mutant and native KPC enzymes of PTV4 strain.

## 4. Discussion

We reported data on four genotypically characterized *Klebsiella pneumoniae* KPC-producing strains resistant to CAZ-AVI, one of which showed a new mutation conferring resistance to CAZ-AVI.

The emergence of *K. pneumoniae* CAZ-AVI-R is a worrisome issue, posing important therapeutic and public health challenges [15,19,20,21,22]. Risk factors for resistance selection have not yet been fully elucidated, however, several authors have identified suboptimal antibiotic exposure as one of the main factors associated with CAZ-AVI-R emergence [13]. In this study, *K. pneumoniae* strains were obtained from patients with a previous long exposure time to CAZ-AVI. However, the development of isolates resistant to CAZ-AVI occurred after 4 days of therapy in one case, therefore, it is difficult to establish the critical duration of exposure to CAZ-AVI associated with resistance. Strains of *K. pneumoniae* resistant to CAZ-AVI have been isolated from anatomical sites other than those of susceptible strains; therefore, it is reasonable to hypothesize an easier selection of resistant CAZ-AVI isolates in anatomical sites less accessible to the antibiotic, where the presence of subtherapeutic drug concentrations is conceivable. However, the limited number of strains studied, and the lack of pharmacokinetic data do not allow definitive conclusions on this issue.

As reported by other studies [20,33,34,35,36,37], restoration of susceptibility to meropenem was noticed in *K. pneumoniae* isolates switching to CAZ-AVI resistance. This phenomenon has been related to amino acid substitutions and conformational changes in the active site of carbapenemase enzymes, that resulted in the restoration of low minimum inhibitory concentrations (MICs) to meropenem. The use of meropenem in these cases was, however, not recommended, as it is often followed by the rapid restoration of resistance to the drug [20,21,37,38].

Our study specifically addresses CAZ-AVI resistance in *Klebsiella pneumoniae* KPC, highlighting the presence of important diagnostic challenges. None of the CAZ-AVI-R strains were detected as KPC-producing strains by the immunochromatographic assay, and they did not grow on a selective chromogenic medium. This behavior, already described by Antonelli and colleagues [29], raises the serious problem of a shortage of diagnostic tools that clinical microbiology laboratories should be aware of. Limitations of the immunochromatographic assay should be considered when testing *Klebsiella pneumoniae* isolates, especially in patients receiving treatment with CAZ-AVI. The rapidity of immunochromatographic tests has important clinical value; hence, the validity of the test should be thoroughly assessed. Accurate epidemiological studies and the identification of risk factors for the development of resistance to CAZ-AVI could help to identify target populations in which use specific molecular assays instead of relying on rapid diagnostic tools. In situations where there is a strong suspicion of possible resistance to CAZ-AVI, genotypic testing in combination with the resistance profile remains the only reliable option for detecting the resistance mechanism, to be confirmed by sequencing the allelic variant.

Finally, in addition to the mutation at position 532, a transition that causes aspartic acid to replace tyrosine (D179Y) known to be associated with CAZ-AVI resistance in KPC, the PTV4 isolate presented another site mutation at position 553. This mutation, never described before, is a transversion from thymine to adenine (CTG-CAG). The mutation causes a KPC variant in which glutamine replaces leucine (Q168L). Observing 3D images of the KPC variant, we can speculate that the amino acid substitution causes a change in the omega loop of KPC-3, which causes resistance to CAZ-AVI. The bla_KPC_ cloning demonstrated that the new mutation was able to confer resistance to CAZ-AVI being the MIC of CAZ-AVI, performed on transformants in a microbroth dilution test, equal to 32 μg/mL.

Continuous reporting of new mutations in the KPC gene is essential, and the identification of risk factors for CAZ-AVI resistance is useful in optimizing the use of the new BLBLI for the treatment of *K. pneumoniae* KPC-producing infections. Microbiologists should be particularly vigilant in monitoring resistance in patients infected and colonized with *K. pneumoniae* KPC-producing strains by using all available diagnostic tools. National studies with a greater number of patients would allow us to better identify the risk factors for CAZ-AVI resistance and address the challenge of emerging MDR pathogens.

## Figures and Tables

**Figure 1 microorganisms-09-02356-f001:**
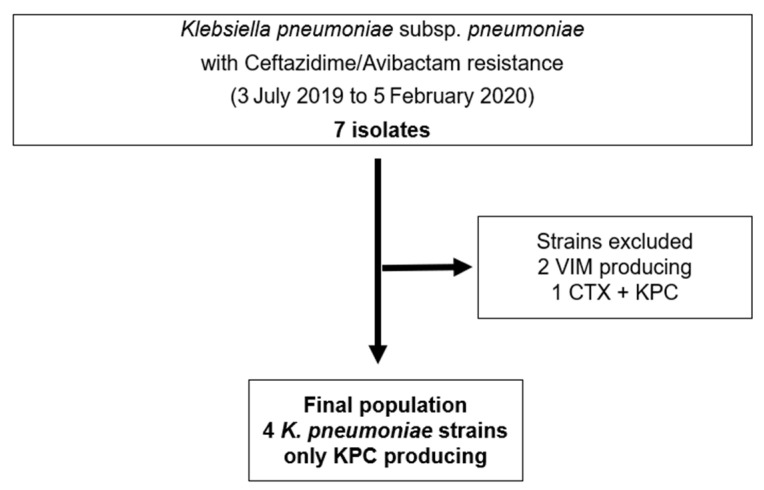
Flow chart showing *K. pneumoniae* CAZ-AVI-R strains selection criteria.

**Figure 2 microorganisms-09-02356-f002:**
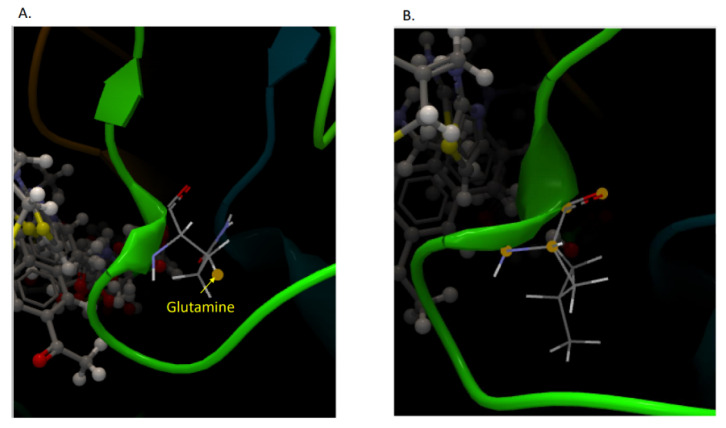
Three-dimensional (3D) images of KPC-3 of PTV4 strain. (**A**) KPC-3 with the mutation in which glutamine replaces leucine (Q168L). (**B**) Wild type. The 3D images have been obtained using a free available bioinformatics web server at https://www.reading.ac.uk/bioinf/servlets/nFOLD/IntFOLD6results.jsp?time=16_5_30_665_7-10-2020_All&md5=295b5b0r3v0mlj1k&targetname=striatin-4#TS (accessed on 13 November 2021).

**Table 1 microorganisms-09-02356-t001:** Details on CAZ-AVI-R and CAZ-AVI-S *Klebsiella pneumoniae* strains.

Isolate ID	Admission Ward	Type of Specimen	Date of Specimen Collection	Meropenem MIC	CAZ-AVI MIC	Time from Isolation of CAZ-AVI-R in Patients with Previous CAZ-AVI-S
PTV1	Vascular surgery	Urine culture	30/04/2019	>16	2	52 days
		Rectal swab	21/06/2019	4	>8	
PTV2	Cardio surgical intensive care	Wound swab (sternotomy)	22/07/2019	>16	≤1	44 days
		Bronchoalveolar lavage	04/09/2019	2	>8	
PTV3	Transplant surgery	Blood culture	15/11/2019	≤0.125	8	1 day
		Intrabdominal abscess	16/11/2019	2	64	
PTV4	Medicine ward	Urine culture	01/02/2020	>64	2	4 days
		Blood culture	05/02/2020	≤0.125	32	

**Table 2 microorganisms-09-02356-t002:** Antimicrobial susceptibilities of the four CAZ-AVI-R strains.

Isolate ID	MICs mg/L												
	AK	CEF	CTZ	CAA	CFT	CIP	COL	GN	IMI	MEM	LEV	PZT	TSU
PTV1	8	>8	>32	>8	>8	>8	≤1	0.5	4	4	>8	>128	>8
PTV2	≤4	>8	>32	>8	>8	>8	≤1	≤0.25	≤1	2	>8	>128	>8
PTV3	≤4	>16	>64	64	>64	>1	1	>8	NA	2	>8	128	>8
PTV4	≤4	>16	>64	32	64	>1	<0.5	>8	NA	≤0.125	4	16	>8

AK: Amikacin; CEF: Cefepime; CTZ: Ceftazidime; CAA: Ceftazidime/Avibactam; CFT: Ceftolozane/Tazobactam; CIP: Ciprofloxacin; COL: Colistin; GN: Gentamycin; IMI: Imipenem; MEM: Meropenem; LEV: Levofloxacin; PZT: Piperacillin/Tazobactam; TSU: Trimethoprim/Sulfamethoxazole; MIC: minimum inhibitory concentration. NA: not available

**Table 3 microorganisms-09-02356-t003:** Details on CAZ-AVI-R strains.

Isolate ID	Type of Specimen	Carbapenemases Detected	Mutation	Amino Acid Substitution
PTV1	Rectal swab	bla _KPC_-KPC3	C814T G523T	H272Y—Histidine vs. Tyrosine D179Y—Aspartic Ac vs. Tyrosine
PTV2	Bronchoalveolar lavage	bla _KPC_-KPC3	C814T G523T	H272Y—Histidine vs. Tyrosine D179Y—Aspartic Ac vs. Tyrosine
PTV3	Intrabdominal abscess drainage	bla _KPC_-KPC3	C814T G523T	H272Y—Histidine vs. Tyrosine D179Y—Aspartic Ac vs. Tyrosine
PTV4	Blood culture	bla _KPC_-KPC3	C814T 553/CTG-CAG	H272Y—Histidine vs. Tyrosine Q168L—Glutamine vs. Leucine

## Data Availability

Analyzed data are available in datasets generated during the study.

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
