# Peer review of "Ceftazidime/Avibactam-Resistant Klebsiella pneumoniae subsp. pneumoniae Isolates in a Tertiary Italian Hospital: Identification of a New Mutation of the Carbapenemase Type 3 (KPC-3) Gene Conferring Ceftazidime/Avibactam Resistance"

_microorganisms, 2021, doi:10.3390/microorganisms9112356_

Round 1

Reviewer 1 Report

The authors have addressed all my concerns and therefore I support publication without further revision.

Author Response

Submission ID microorganisms-1465691

Title - Ceftazidime/avibactam-resistant Klebsiella pneumoniae subsp. pneumoniae isolates in a tertiary Italian hospital: identification of a new mutation of the carbapenemase type 3 (KPC-3) gene conferring ceftazidime/avibactam resistance

Reviewer 1

The authors have addressed all my concerns and therefore I support publication without further revision.

Answer: thank you for your reply and acceptance of the article

Reviewer 2 Report

The revised manuscript well addressed the issues indicated previously and this time asks the authors to consider the several minor points.

Keywords: Q168L substitution (instead of mutation)

line 42: New-Delhi

line 73: a new amino acid substitution (instead of mutation)

line 91:imipenemase (not imipenase)

line 115: Q168L substitution (instead of mutation)

Fig. 1: Klebsiella pneumoniae subsp. pneumoniae (not subspecie)

Table 1 and 2: CAZ-AVI MIC value for PTV1 is >8 in Table 1 while 16 in Table 2. Which is correct?

Table 3: amino acid substitution ( not aminoacyl)

lines 164 and 223: 32 mg/L (not μg/L)

line 223: transformants L, what does "L" indicate?

There are some type errors through the manuscript. The authors should reconfirm them before submission.

Author Response

Submission ID microorganisms-1465691

Title - Ceftazidime/avibactam-resistant Klebsiella pneumoniae subsp. pneumoniae isolates in a tertiary Italian hospital: identification of a new mutation of the carbapenemase type 3 (KPC-3) gene conferring ceftazidime/avibactam resistance

Reviewer 2

Comments and Suggestions for Authors

The revised manuscript well addressed the issues indicated previously and this time asks the authors to consider the several minor points.

Keywords: Q168L substitution (instead of mutation)

Answer: modified according to reviewer’s suggestions

line 42: New-Delhi

Answer: modified according to reviewer’s suggestions

line 73: a new amino acid substitution (instead of mutation)

Answer: modified according to reviewer’s suggestions

line 91:imipenemase (not imipenase)

Answer: done

line 115: Q168L substitution (instead of mutation)

Answer: done

Fig. 1: Klebsiella pneumoniae subsp. pneumoniae (not subspecie)

Answer: done

Table 1 and 2: CAZ-AVI MIC value for PTV1 is >8 in Table 1 while 16 in Table 2. Which is correct?

Answer

We apologize with the reviewer, currently, PTV1 resistance to CAZ-AVI results > 8 both in table 1 and 2 (pag 8 line 141 and 143)

Table 3: amino acid substitution ( not aminoacyl)

Answer: done

lines 164 and 223: 32 mg/L (not μg/L)

Answer:

We apologize with the reviewer, the mistake has been corrected and 32 mg/ml now in present at page 5 line 165 and at page 6, line 224.

line 223: transformants L, what does "L" indicate?

Answer: we apologize for the mistake,’ L’ has been removed

There are some type errors through the manuscript. The authors should reconfirm them before submission.

Answer: we thanks to the reviewer for the suggestions, the article has been carefully re-read and all typos corrected.

All changes to the manuscript are highlighted in the text

This manuscript is a resubmission of an earlier submission. The following is a list of the peer review reports and author responses from that submission.

Round 1

Reviewer 1 Report

In this manuscript, Fontana et al characterized the ceftazidime/avibactam-resistant Klebsiella pneumoniae isolates and identified an novel KPC-3 mutation. The topic is important and interesting. I have few comments as described below,

Major comment:

Although the authors identified an novel KPC mutation (Q168L), however, they only predicted and compared the 3D structure of mutant and naïve KPC enzymes. Moreover, in Table 2, PTV3 (KPC H272Y) showed higher CAA resistance, compared to PTV4 (KPC H272Y and Q168L). Therefore, the authors should further characterize the effect of novel KPC mutation.

Minor comments:

  1. P2. "blaKPC" gene.
  2. P6. "...... be aware of. Limitations of the immunochromatographic assay  ....." should be "...... be aware of the limitations of immunochromatographic assay  ....."

Reviewer 2 Report

This manuscript by Fontana et al. reported the molecular characteristics of CAZ/AVI-resistant Klebsiella pneumoniae subsp. pneumoniae isolates from a university hospital, Italy and identified two amino acid substitutions, one is already-known H272Y and the other is a new one, Q168L, in KPC carbapenemases. In addition, the authors identified that these CAZ/AVI resistant isolates emerged from CAZ/AVI susceptible strains in patients. Overall, the manuscript was clearly written and the methodology employed was straightforward. However, there are some points that I can't understand and there seems to lack experimental evidences leading to the author's conclusion.

 1) Although the H272Y amino acid substitution leads to the change from KPC-2 to KPC-3, this substitution alone is not involved in CAZ/AVI resistance. To acquire CAZ/AVI resistance, further changes such as amino acid substitutions (V240G, T243A, D179Y, and T243M) and amino acids insertion/deletion (see. Hobson et al. Clinical Microbiology and Infection 27(2021) 1172.e7-e10) are necessary in KPC carbapenemases. However, these additional changes to confer CAZ/AVI resistance have not been identified in KPC carbapenemases in this study. The H272Y substitution alone could not explain CAZ/AVI resistance. The author's conclusions in the present study are inconsistent with those described in the past relevant publications. Please explain.

2) The authors concluded that the newly identified Q168L in KPC was involved in CAZ/AVI resistance. However, this remains to be uncertain. The authors should perform the experiments such as site-directed mutagenesis to clear the role of Q168L amino acid substitution in KPC.

3) The authors concluded that CAZ/AVI resistant isolates were switched from CAZ/AVI susceptible strains. The authors must reveal that CAZ/AVI resistant isolates and CAZ/AVI susceptible strains were genetically clonal though the experiments such as Sequence Typing and PFGE.

Reviewer 3 Report

The communication of Fontana et al. is interesting but needs improvements.

Title:

  • Genomic: readers expect that the genome sequences have been analyzed and therefore the word genomic is misleading
  • subsp. : should not be written in italic
  • The authors detected a mutation in KPC-gene, but there are no genetic experiments in the manuscript that would proof that this is related to ceftazidine/avibactam resistance
  • KPC-3: there is no explanation in the manuscript what does it mean number 3

There are no line numbers in the manuscript.

Abstract:

- First sentence of the abstract: needs a reference

- Subsp.: not in italic

Introduction:

  • Klebsiella pneumoniae: in italic
  • subsp: not in italic
  • spp.: not in italic
  • blaKPC isolates: bla KPC genes?
  • Last sentence: there are no genetic experiments which would proof that these mutations confer resistance to CAZ-AVI.

Materials and methods:

  • subsp.: not in italic
  • 2.2.: please, add a brief explanation of the method of Favaro et al.
  • 2.4.: reference Goldfarb should be added in number

Results:

  • No sequence can be fetched with acc. no. MT939316
  • Figure 2: which program has been used for inferring this structure?

Spelling throughout the manuscript should be checked.

Reviewer 4 Report

I suggest that this manuscript would be suitable for publication in Microorganisms following minor revisions to the text as follows:

Page 1.

In the title change ‘Hospital’ to ‘hospital’

Abstract, line 11.  Change ‘resulted’ to ‘they were’

Abstract, line 13.  Rewrite ‘for the prompt detection of Klebsiella pneumoniae KPC-producing’

Add further keywords.

Introduction line 1.  Change ‘The antimicrobial’ to ‘Antimicrobial’

Introduction line 12.  Italicise ‘Klebsiella pneumoniae’

Page 2

Materials and Methods, line 5.  Change ‘mechanism’ to ‘mechanisms’

Materials and Methods, line 18.  Remove ‘the’

Materials and Methods, line 20.  Insert ‘the’ before ‘manufacturer’s’

Page 3

Figure 1 shows a ‘Flow chart’ rather than an ‘Algorithm’

Results, second paragraph, line 4.  Change ‘Time’ to ‘The length of time’

Page 4

Second paragraph, line 6.  Insert ‘the’ before ‘NCBI’

Second paragraph, line 7.  Change ‘naïve’ to ‘native’

Page 5

Figure 2 legend.  Change ‘wild type’ to ‘Wild-type’

Fourth paragraph, line 12.  Change ‘in which’ to ‘that’

Page 6

Under Funding it says ‘This research received no external funding’ then under Conflicts of Interest it says ‘research funding from Gilead’

If there are no aknowledgements, remove ‘Acknowledgments:’

Conflicts of Interest, line 1.  Gilead is listed twice.

Conflicts of Interest, line 1.  Insert ‘and’ before ‘Pfizer’

Conflicts of Interest, line 2.  Insert ‘and’ before ‘Abbvie’